# Combination Immunotherapies to Overcome Intrinsic Resistance to Checkpoint Blockade in Microsatellite Stable Colorectal Cancer

**DOI:** 10.3390/cancers13194906

**Published:** 2021-09-29

**Authors:** Chang Woo Kim, Hong Jae Chon, Chan Kim

**Affiliations:** 1Department of Surgery, Ajou University School of Medicine, 164 World Cup-ro, Yeongtong-gu, Suwon 16499, Korea; gkim@aumc.ac.kr; 2Medical Oncology, CHA Bundang Medical Center, CHA University School of Medicine, 59 Yatap-ro, Bundang-gu, Seongnam 13496, Korea

**Keywords:** colorectal cancer, microsatellite stability, mismatch repair proficiency, immunotherapy, immune checkpoint inhibitor, resistance

## Abstract

**Simple Summary:**

Immune checkpoint inhibitors have revolutionized the treatment landscape of microsatellite instable colorectal cancer. However, their efficacy is very limited in patients with microsatellite stable colorectal cancer. Recent preclinical and clinical studies have suggested the involvement of various tumor cell-intrinsic and tumor microenvironmental mechanisms in primary resistance to immunotherapy in microsatellite stable colorectal cancer. Because microsatellite stable colorectal cancers are immunologically heterogeneous with different immune evasive mechanisms, it is critical to precisely identify the major mechanism of resistance in each patient and to accordingly apply personalized combination immunotherapy strategies.

**Abstract:**

Although immune checkpoint inhibitors (ICIs) have shown promising results in the treatment of treating various malignancies, progress has been severely limited in metastatic colorectal cancer (mCRC). ICIs are effective in a fraction of patients with microsatellite instability-high mCRC but have little clinical efficacy in patients with microsatellite stable (MSS) mCRC, which accounts for 95% of mCRC cases. MSS mCRCs are considered to have intrinsic resistance to ICI monotherapy through multiple mechanisms. (1) They are poorly immunogenic because of their low tumor mutation burden; (2) frequent activation of the WNT/β-catenin signaling pathway excludes intratumoral CD8^+^ T cell immunity; (3) the tumor microenvironment is immunosuppressive because of the presence of various immunosuppressive cells, including tumor-associated macrophages and regulatory T cells; and (4) frequent liver metastasis in MSS mCRC may reduce the efficacy of ICIs. To overcome these resistance mechanisms, combination approaches using various agents, including STING agonists, MEK inhibitors, VEGF/R inhibitors, WNT/β-catenin inhibitors, oncolytic viruses, and chemo/radiotherapy, are actively ongoing. Preliminary evidence of the efficacy of some has been shown in early clinical trials. This review summarizes novel combination immunotherapy strategies described in recent preclinical and clinical studies to overcome the limitations of ICI monotherapy in MSS mCRC.

## 1. Introduction

Immunotherapy has changed the paradigm of treatment for advanced malignancies. James P. Allison and Tasuku Honjo won the Nobel Prize for Medicine in 2018 in recognition of their efforts to identify programmed cell death-1 (PD-1) and cytotoxic T-lymphocyte associated protein 4 (CTLA-4) [1,2,3]. As representative agents for current cancer immunotherapy, immune checkpoint inhibitors (ICIs) have distinct mechanisms of action compared to traditional cytotoxic chemotherapy. They selectively block PD-1/PD-L1 or CTLA-4/CD80 or CD86 interactions and subsequently enhance the ability of immune cells to attack cancer cells and establish long-lasting anti-tumor immunity, whereas traditional cytotoxic chemotherapeutic agents directly disturb cancer cell activity or viability regardless of the genetic or epigenetic background of the tumor or tumor microenvironment (TME) [4,5,6,7].

Numerous preclinical and clinical trials have demonstrated that ICIs have dramatic anti-tumor efficacies and reduced toxicities. They are effective for multiple types of tumors, including melanoma, lung, kidney, urothelial, gastric, cervical, and head and neck cancer [8,9,10,11,12]. Moreover, various adverse events experienced by most patients following treatment with cytotoxic or targeted agents rarely occur in patients who are treated with ICIs, although immune-related adverse events, including autoimmune diseases, have been observed. Although ICIs were initially applied to treat advanced metastatic diseases, they are now being evaluated in neoadjuvant or combination settings [13,14,15].

However, the objective response rates (ORRs) of ICIs vary from 15% to 50% among different types of cancer, showing unsatisfactory results [3,16,17,18]. Colorectal cancer (CRC), which is among the most common causes of cancer deaths worldwide, is a representative ‘non-responder’ to ICIs [19,20,21]. In patients newly diagnosed with metastatic CRC (mCRC), only approximately 5% of mCRC cases are microsatellite instability-high/deficient mismatch repair (MSI-H/dMMR) [22]. The mutation or hypermethylation of MMR genes, such as *MLH1*, *MSH2*, *MSH6*, and *PMS2*, results in failure to repair replication-associated errors in their DNA and MSI-H [23,24]. The accumulation of DNA abnormalities in MSI-H/dMMR tumors can lead to more frequent somatic mutations and tumor neoantigens, and MSI-H tumors tend to respond more favorably to ICIs. The first phase I trial of nivolumab, a PD-1 antibody, achieved only one complete response among 14 patients with mCRC and the case was a tumor with dMMR with PD-L1^+^ lymphocyte infiltration surrounding tumor cells [25]. Subsequent studies showed consistent results: mCRC with MSS/proficient MMR (pMMR) did not respond to ICIs, whereas MSI-H/dMMR mCRC responded favorably to ICIs [22,26,27,28]. In a phase II trial of pembrolizumab in patients with mCRC, the ORR was 40% in patients with MSI-H/dMMR mCRC, whereas no patients with MSS/pMMR showed an objective response [29]. To overcome the limited efficacy of ICI monotherapy, many investigators have attempted various combination immunotherapeutic approaches in patients with MSS/pMMR mCRC (Table 1). Here, we review the possible resistance mechanisms of immunotherapy and therapeutic targets for combination immunotherapy in MSS/pMMR mCRCs.

## 2. Mechanisms of Intrinsic Resistance to Immune Checkpoint Blockade in MSS/pMMR mCRC

Various tumor cell-intrinsic and microenvironmental factors mediate intrinsic resistance to immune checkpoint blockade in MSS/pMMR mCRC (Figure 1).

### 2.1. Low Tumor Mutation Burden and Lack of Tumor Antigen

Cancer is primarily a consequence of accumulated genetic mutations [39]. Mutations that can modify the coding sequence of amino acids (non-synonymous mutations) generate novel peptides that can be recognized as non-self by the immune system and act as tumor neoantigens [40,41]. Therefore, the number of somatic mutations within a given tumor—known as the tumor mutation burden (TMB)—is approximately proportional to the number of tumor neoantigens [41,42]. Moreover, recent reports suggested that tumors with a high TMB are associated with better clinical responses to immunotherapy than those with a low TMB, as TMB high tumors presumably have more tumor neoantigens and thus are more likely to be recognized and attacked by the immune system [40,41,43]. Interestingly, the average TMB of MSI-H/dMMR tumors is reported to be very high (30 mutations/MB), as their mismatch repair machinery is typically deficient. In contrast, the average TMB of MSS/pMMR tumors is only four mutations/MB, indicating lower immunogenicity compared to MSI-H tumors [44]. These results suggest that MSS CRCs have fewer tumor neoantigens than MSI-H CRCs; therefore, the immunologic response to ICIs may be much weaker in MSS CRCs. However, as TMB-low tumors such as liver and kidney cancers respond to ICI treatment, a low TMB alone cannot fully explain why MSS CRC is refractory to ICI therapy, indicating that other mechanisms of resistance also exist [21].

### 2.2. Active WNT/β-catenin Signaling and Immune Exclusion

CD8^+^ T cells are the primary factors in anti-tumor adaptive immunity [45]. Therefore, intratumoral CD8^+^ T cell infiltration has been correlated with the response to ICIs in numerous preclinical and clinical studies [18]. WNT/β-catenin signaling is known to be correlated with the exclusion of T cells within the TME in various solid malignancies [21,46]. WNT/β-catenin signaling activation is frequently observed in MSS CRC, whereas it is rarely observed in MSI-H CRC [46,47]. Approximately 50% of patients with colon cancer displayed non-T-cell-inflamed tumors in RNA-seq analysis, and β-catenin activation was very frequent (98.9%) in their tumor tissue [46]. In a preclinical melanoma model, the activation of oncogenic WNT/β-catenin signaling represses the recruitment of CD103^+^ dendritic cells, thereby hindering T cell activation [48]. β-catenin also suppressed the transcription of T-cell recruiting chemokine, CCL4, from dendritic cells and thus reduced the infiltration of CD8^+^ T cells within the TME in colon cancer and melanoma [48,49]. Recent reports described that a pharmacological inhibition of β-catenin could reactivate cancer immunity by increasing dendritic cells, upregulating CCL4, and promoting the intratumoral infiltration of CD8^+^ T cells in various tumor models, including colon cancer [49,50,51]. Moreover, β-catenin inhibition could strengthen the efficacy of PD-1 immune checkpoint blockade in multiple tumor models such as lung, breast, kidney, liver, and brain tumors [50,51,52,53]. There is also crosstalk between WNT/β-catenin signaling and tumor-associated macrophages. Colon cancer cells promote IL-β secretion from macrophages. IL-β can prevent β-catenin degradation in colon cancer cells by phosphorylating GSK-3β [54,55]. Overall, active β-catenin signaling excludes T cells in the TME and confers resistance to immune checkpoint blockade.

### 2.3. VEGF-driven Immunosuppressive TME

Vascular endothelial growth factor (VEGF) plays a crucial role in tumor angiogenesis [56,57]. Upregulation of the VEGF/VEGF receptor (VEGFR) signaling pathway is frequently observed in CRC and promotes the progression and metastasis of colon cancer [58]. Moreover, VEGF is a major immunosuppressive factor that mediates immune evasive mechanisms at multiple stages of the cancer–immunity cycle. It not only suppresses the activation and maturation of dendritic cells, but also disturbs the activation and effector functions of T cells [18]. Moreover, it accumulates immunosuppressive cells such as myeloid-derived suppressor cells, tumor-associated macrophages (TAMs), and regulatory T cells (Tregs) within the TME [59]. VEGF-driven Tregs promote M2-like TAM activity by inhibiting the secretion of IFN-γ from CD8^+^ T cells [60]. Subsequently, M2-like TAMs further produce excess VEGF within CRC, generating a feed-forward loop of immunosuppression within the tumor [21]. Therefore, VEGF inhibition can block this feed-forward loop and normalize the tumor immune microenvironment to enhance the efficacy of ICI therapy, in addition to its traditional anti-angiogenic role as a targeted agent in mCRC.

### 2.4. TGF-β Signaling Activation

Transforming growth factor (TGF)-β signaling is another immune evasive mechanism in CRC. Although MSI-H CRCs do not show dominant TGF-β signaling, a subtype of MSS CRC, mesenchymal type CRC, frequently shows an activation of TGF-β signaling [61]. TGF-β enhances cancer-associated fibroblasts to promote intratumoral fibrosis and suppresses anti-tumor immunity [62,63]. Therefore, TGF-β is partially involved in the resistance to immunotherapy in MSS CRC.

### 2.5. Immune Tolerance by Liver Metastasis

The liver is the most common site of distant metastasis in mCRC and is a central organ of immune tolerance [17,64]. Various mechanisms of immune tolerance in the liver have been reported. An incomplete activation of CD8^+^ T cells and poor CD4^+^ T cells within the liver may lead to exhaustion or premature death of T cells. In addition, intrahepatic Kupffer cells, myeloid-derived suppressor cells, and dendritic cells can promote Tregs while suppressing effector T cells [65,66,67,68]. In patients with advanced solid tumors treated with various immuno-oncologic agents in phase I trials, in which the most common histologies were melanoma and gastrointestinal cancers, liver metastases were associated with worse overall survival than the absence of liver metastases (median 8.1 vs. 21.9 months, *p* = 0.017) [69]. Other studies have shown that patients with liver metastasis gain a reduced benefit from ICIs independent of other biomarkers in patients with melanoma and non-small cell lung cancer [70,71,72]. Consistently, patients with advanced gastric cancer with hyperprogression of disease (HPD) after nivolumab treatment display more frequent liver metastases than those without HPD (77 vs. 41%, *p* = 0.029) [73]. A recent study showed that liver metastases siphon activated CD8^+^ T cells from the systemic circulation and that hepatic myeloid cells induce T cell apoptosis through the Fas-FasL signaling pathway in preclinical models of colon cancer with liver metastasis, thus generating a systemic immune desert [68]. The findings of another study suggested that liver metastasis induces tumor-specific immune suppression in distant tumors by activating Tregs and modulating CD11b^+^ monocytes in preclinical models of colon cancer [67]. Intriguingly, patients with liver metastases displayed reduced diversity and function of intratumoral T cells and decreased T cell signature scores [68].

Patients with MSS mCRC more frequently showed liver metastasis compared to patients with MSI-H mCRC (71.0% vs. 26.7, *p* = 0.001) [74]. Therefore, liver metastasis may be a cause of refractoriness to ICI therapy in MSS mCRC. A recent study by Want et al. supports this possibility. Among 95 patients with MSS mCRC treated with PD-1/PD-L1 targeted therapy, the overall ORR was 8.4%, where patients without liver metastasis achieved an ORR of 19.5% and those with liver metastases showed no response [75]. Moreover, patients without liver metastasis had superior progression-free survival compared with those with liver metastasis (4.0 vs. 1.5 months, *p* < 0.001). Moreover, multivariate analysis indicated that liver metastasis was one of the most important predictors of faster progression after PD-1/PD-L1 blockade, even after adjusting for other variables such as the primary tumor size, RAS status, BRAF status, and TMB [75]. As some patients with MSS mCRC who do not have liver metastases favorably respond to ICIs, it is necessary to thoroughly investigate the possibility of immune checkpoint blockade in this subset of patients.

Overall, a low TMB and poor immunogenicity, WNT/β-catenin signaling and immune exclusion, VEGF-driven immunosuppression, TGF-β signaling, and immune tolerance induced by liver metastasis may result in resistance to immunotherapy in MSS/pMMR mCRC.

## 3. Overcoming Resistance to Immunotherapy in MSS/pMMR mCRC with Novel Combination Strategies

### 3.1. MEK Inhibitor

Because most MSS mCRCs are poorly immunogenic, studies have been performed to overcome this limitation by enhancing immunogenicity using various agents. The mitogen-activated protein kinase pathway is known to modulate the expression of major histocompatibility complex class I molecules, thus regulating antigen presentation during carcinogenesis [21,76]. Mitogen-activated protein/extracellular signal-regulated kinase (MEK) inhibition enhances major histocompatibility complex class I molecule expression and intratumoral T cell infiltration in preclinical CRC models [77,78]. A phase Ib trial combining atezolizumab (ant anti-PD-L1 antibody) and cobimetinib (a MEK inhibitor) reported a response rate of 8% (7 of 84 patients with mCRC); 6 of these patients had MSS/pMMR mCRC [37]. Subsequently, a randomized, open-label, phase III-controlled trial (IMblaze 370) evaluated the clinical efficacy of three treatment groups in mCRC: atezolizumab with cobimetinib, atezolizumab alone, and regorafenib alone, in 93%, 92%, and 89% of MSS/pMMR mCRC patients, respectively [38]. However, the experimental arm failed to show a significant improvement in overall survival, which was the primary endpoint of the trial.

### 3.2. Stimulator of Interferon Genes Agonist

Stimulator of interferon (IFN) genes (STING) is an innate immune sensor that links innate and adaptive immunity during the cancer–immunity cycle [20,79]. STING signaling is critical for type-I IFN responses [20,80]. STING activation triggers robust type-I IFN secretion from dendritic cells and other stromal cells, stimulates the cross-priming of tumor neoantigens to CD8^+^ T cells, and finally induces anti-tumor adaptive immunity within the TME [18,81,82]. A previous study showed that intratumoral STING treatment suppressed tumor growth and improved survival in a preclinical model of CRC [20,82,83]. The experimental group showed marked increases in CD8^+^ cytotoxic T cells and IFN-γ after STING treatment. Moreover, given that repeated STING agonist injection upregulated intratumoral PD-L1 expression, the combination of STING agonist and anti-PD-1 antibody is expected to show synergism in anti-tumor immune responses and tumor control. However, no clinical data assessing the safety and efficacy of STING agonists in patients with CRC have been reported.

### 3.3. Cytotoxic Chemotherapy

Chemotherapy may also be a reasonable option for enhancing cancer immunotherapy. Current chemotherapy regimens for mCRC have been established and include various agents, such as platinum analogs (oxaliplatin), topoisomerase inhibitors (irinotecan), epidermal growth factor receptor (EGFR) inhibitors (cetuximab and panitumumab), and VEGF inhibitors (bevacizumab and aflibercept) [84,85,86]. The addition of these agents to ICIs has been attempted in preclinical and clinical studies of CRC. Interestingly, cytotoxic chemotherapy agents such as 5-fluorouracil and oxaliplatin enhance the immunogenicity of CRC by promoting the release of tumor antigens from tumor cells and their uptake by dendritic cells. 5-Fluorouracil, a historical anti-metabolite for CRC, promotes T-cell dependent antitumor responses in vivo by stimulating immunogenic cell death and by selectively killing immunosuppressive myeloid cells [87,88]. Oxaliplatin not only stimulates pro-apoptotic calreticulin exposure but also upregulates PD-L1 on dendritic cells, which may be related to the response to ICIs [89,90]. MEDITREME is a phase Ib/II trial that aimed to assess the efficacy and safety of FOLFOX (folinic acid, fluorouracil, and oxaliplatin) with durvalumab and tremelimumab for mCRC [91]. An interim analysis of this study showed six-month progression-free survival rates of 62.5% after a median treatment duration of 13.4 months. Moreover, most adverse events were well-tolerated and related to FOLFOX, although some immune-related adverse events, such as thyroid dysfunction and hypophysitis, were related to tremelimumab. A single-arm phase II trial (NIVACOR) to assess the efficacy of nivolumab with FOLFOXIRI (fluorouracil, leucovorin, oxaliplatin, and irinotecan) plus bevacizumab is currently recruiting patients [92]. Preliminary safety results showed that this combination regimen was generally tolerable, with acceptable toxicity profiles. CheckMate 9X8 is another ongoing randomized phase II/III trial comparing the standard chemotherapy plus nivolumab versus standard chemotherapy in treatment-naïve mCRC [93]. The final results of these studies will validate the effectiveness of combining ICIs with cytotoxic chemotherapy in MSS mCRC.

### 3.4. VEGF/R Inhibitor

REGONIVO, a phase Ib trial of regorafenib (a VEGFR2 inhibitor) plus nivolumab showed promising results in Japanese patients with mCRC and metastatic gastric cancer [34]. Among the 25 patients with mCRC, only one had MSI-H/dMMR mCRC. The response rate in MSS/pMMR mCRC after excluding patients with MSI-H/dMMR mCRC was 33.3%, and adverse events were consistent with the known toxicity profiles of the two agents. Based on this result, a single-arm phase II study of regorafenib and nivolumab was performed in North America (NCT0412733) [94]. Disappointingly, this phase II study showed a response rate of only 7%, which is inconsistent with the Japanese data. A detailed analysis of this study showed that patients without liver metastasis benefited more with this combination therapy with a 22% response rate, whereas those with liver metastasis showed no response. Therefore, this combination regimen requires further validation in a subset of patients with mCRC without liver metastasis, rather than an all-comer. Moreover, further studies are warranted to determine the impact of liver metastasis on the efficacy of immune checkpoint blockade in mCRC. BACCI, a randomized, double-blind, phase II-controlled trial that compared capecitabine/bevacizumab/atezolizumab with capecitabine/bevacizumab/placebo in patients with refractory mCRC, yielded an ORR of 8.54% in the triple combination group compared to 4.35% in the dual combination group [33]. Using a similar concept, Ren et al. designed a phase II trial to determine the toxicity and efficacy of SHR-1210 (a monoclonal anti-PD-1 antibody) plus apatinib (VEGFR inhibitor) in MSS/pMMR mCRC [95]. However, they reported that the ORR was 0%, with intolerable toxicities. Unlike the previous two trials that targeted PD-1, an anti-PD-L1 (avelumab) was combined with regorafenib in a phase II REGOMUNE trial [96]. They reported that the best response was stable disease in 53.5% of patients, whereas there was no objective response in patients with MSS/pMMR mCRC.

### 3.5. WNT/β-catenin Signaling Inhibitors

The activation of WNT/β-catenin signaling is frequently observed and is associated with T-cell exclusion in MSS/pMMR CRC [21,46]. Preclinical studies revealed that targeting WNT/β-catenin enhanced tumor antigen release, dendritic cell activation, T-cell cross-priming, and intratumoral infiltration of T cells into the tumor [97]. WNT inhibitors are mostly used in the very early stages of clinical development, and their clinical efficacy is not well established in mCRC. Early clinical trials of WNT inhibitors, LGK974 and CGX1321, in combination with ICIs are currently ongoing, and the results are expected to be announced soon. Although data on WNT inhibitors are limited in mCRC, there is evidence for combination of WNT inhibitors with ICIs in gastrointestinal tumors. DKN-01 is a monoclonal antibody against DKK1 that modulates Wnt signaling. In a phase Ib study of DNK-01 in combination with pembrolizumab in patients with gastroesophageal cancers, the combination therapy resulted in 6-week disease control rates of 75% and ORR of 11% without significant toxicities. Because DKK1 is commonly overexpressed and WNT signaling is frequently activated in CRC, the combination of DKN-01 and anti-PD1 therapy may be clinically applicable in MSS mCRC [98].

### 3.6. Anti-CTLA-4 Antibody

CTLA-4 was the first inhibitory immune checkpoint to be identified. Early results from preclinical studies revealed the effects of CTLA-4 blockade on effector T cells, whereas recent reports suggested a role for CTLA-4 in depleting Tregs [2,99,100,101]. However, the role of anti-CTLA-4 antibodies in Treg depletion remains controversial. Tang et al. proposed that anti-human CTLA-4 antibodies induce tumor rejection by selective depletion of Tregs within tumors rather than by blocking the B7-CTLA-4 interaction in lymphoid organs [102]. In contrast, Sharma et al. suggested that the currently used anti-CTLA-4 antibodies ipilimumab and tremelimumab do not change or deplete FoxP3^+^ Tregs in human tumors [99]. They also suggested that the efficacy of anti-CTLA-4 antibody could be enhanced by engineering the Fc portions of current antibodies to promote Fc-mediated depletion of intratumoral Tregs. Although the exact mechanisms of action of CTLA-4 blockade remain unclear, the addition of anti-CTLA-4 to anti-PD-1 antibody demonstrated superior anti-tumor effects in various malignancies, including melanoma, kidney, and lung cancer [26]. Although the CheckMate 142 trial showed increased efficacy of the nivolumab and ipilimumab combination for MSI-H/dMMR mCRC, there was no clinical response in patients with MSS/pMMR mCRC [22]. Other investigators compared the combination treatment group with durvalumab (anti-PD-L1) and tremelimumab (anti-CTLA-4) and the best supportive care group in a phase II randomized trial (CCTG CO.26 trial) in patients with refractory mCRC. In this trial, none of the patients had dMMR tumors. With a median follow-up duration of 15.2 months, the median overall survival was 6.6 months for durvalumab and tremelimuab and 4.1 months for the best supportive care group (*p* = 0.07). Moreover, patients in the combination group showed a higher disease control rate than those in the best supportive care group (22.7% vs. 6.6%, *p* = 0.006) [103]. Overall, there is limited evidence supporting the use of current-generation CTLA-4 antibodies in MSS/pMMR mCRC. However, next-generation CTLA-4 antibodies that can deplete intratumoral Tregs are expected to overcome immunotherapy resistance in patients with mCRC.

### 3.7. Oncolytic Virotherapy

Oncolytic viruses are emerging candidates for enhancing the efficacy of ICIs in MSS CRC. Oncolytic viruses induce direct destruction of tumors by selective infection and lysis of tumor cells, followed by immunogenic cell death that occurs with the release of tumor-associated antigens [104,105]. Pexa-Vec (JX-594) is a genetically engineered oncolytic vaccinia virus in which the thymidine kinase (TK) gene has been deleted and the gene encoding human granulocyte macrophage colony-stimulating factor (GM-CSF) inserted [106,107,108]. Because it lacks TK, it can replicate only in TK-overexpressing cancer cells, thus showing high selectivity to cancer cells. Moreover, GM-CSF upregulation by Pexa-Vec facilitates the maturation and activation of dendritic cells within the TME, which enhances the cross-presentation of tumor-antigens and activation of T cells [104,106,108,109]. In a preclinical model of immunotherapy-resistant colon cancer, Pexa-Vec enhanced anti-tumor immunity by remodeling the tumor immune microenvironment [110]. Pexa-Vec selectively killed tumor cells and restored the effector function of tumor-infiltrating T cells. Moreover, Pexa-Vec synergized with anti-PD-1 ICIs to further suppress peritoneal dissemination of colon cancer and eliminate malignant ascites within the peritoneal cavity. A phase I/II study of Pexa-Vec in combination with durvalumab vs. tremelimumab alone for MSS mCRC is currently ongoing, and preliminary interim analysis revealed that this combination is well tolerated without new safety concerns, although long-term efficacy results have not been reported [111]. Reovirus, an orally administered oncolytic virus, also enhanced anti-tumor immunity in several preclinical studies [112,113,114]. In a preclinical study, the oral reovirus RC402 elicited potent and long-lasting anti-tumor immunity in distant tumors by enhancing the infiltration of CD8^+^ cytotoxic T cells and reducing Tregs. Moreover, RC402 cooperated with PD1 blockade to completely regress colon cancer and prolong the survival of tumor-bearing mice [114]. Parakrama et al. also reported that an intravenous administration of reovirus to patients with mCRC resulted in antigen-presenting cell stimulation and CD8^+^ T cell activation, which can be used as an immunomodulator [115].

### 3.8. Liver-Directed Radiotherapy

Radiotherapy is an important treatment modality that can ablate or control liver metastasis in patients with mCRC. Because radiotherapy can induce lethal DNA damage and immunogenic cell death in irradiated tumor cells, this treatment has traditionally been considered to modulate anti-tumor immunity. Recent preclinical studies have suggested that radiotherapy can trigger the type I IFN response through the cGAS-STING signaling pathway, leading to the activation of anti-tumor T cell immunity [116]. The immune-activating effects of radiotherapy depend on the radiation dose and schedule. Although three consecutive doses of 8 Gy stimulate innate immunity through cGAS-STING activation, a single dose of 20 Gy abrogated radiotherapy-induced innate immunity [117]. Consistent with these immunogenic effects, radiation has already shown good synergism with ICIs in various malignancies, suggesting that it is an effective immune enhancer [118]. In a recent study, Yu et al. reported that liver-directed immunotherapy can remodel the TME of the liver and thus overcome immunotherapy resistance induced by liver metastasis in a preclinical model of colon cancer. Moreover, liver-directed immunotherapy can enhance the efficacy of anti-PD-L1 therapy in distant tumors beyond the liver [68]. A non-randomized phase II trial assessed the efficacy of pembrolizumab in combination with radiotherapy in patients with MSS/pMMR mCRC. In the interim analysis, the combination group showed a 9% ORR (1 of 11 patients) [119]. In mCRC, a pilot study combining an anti-PD1 fusion protein (AMP-224) with low-dose cyclophosphamide and stereotactic radiotherapy was performed in patients who were refractory to standard chemotherapy. This combination therapy was well tolerated and showed a 20% disease control rate, a median progression-free survival of 2.8 months, and a median overall survival of 6.0 months [120]. Overall, frequent liver metastasis may explain immunotherapy refractoriness in MSS/pMMR mCRCs, which may be overcome by using local immune-modulating therapies such as liver-directed radiotherapy.

## 4. Conclusions

Immunotherapy has revolutionized the treatment of MSI-H/dMMR mCRC in the last five years; however, 95% of patients with mCRC have the MSS subtype and thus have been left out of this breakthrough. MSS mCRC is refractory to ICI therapy because of the lack of tumor antigens, exclusion of T cells by WNT/β-catenin activation, and VEGF-driven immunosuppressive TME. Moreover, frequent liver metastasis in MSS mCRC may siphon anti-tumor T cells and attenuate the efficacy of ICIs. Attempts to overcome the resistance of MSS CRC to ICIs are currently ongoing with the combined administration of various therapeutics targeting MEK, STING, VEGFR, or WNT, or using oncolytic viruses. Some of early clinical trials have revealed preliminary efficacy signals. However, the data are immature and need further validation. Because MSS mCRC is a cluster of immunologically heterogeneous tumors rather than a single disease entity, it is critical to precisely identify the main mechanism of resistance to immunotherapy in each patient and to apply personalized immunotherapeutic approaches accordingly. For example, in T-cell non-inflamed cold tumors, innate immune agonists such as STING agonists or oncolytic viruses may be effective in enhancing tumor immunogenicity and promoting intratumoral T cell infiltration. In addition, for cancers with very potent WNT/β-catenin activity, it may be better to use Wnt/β-catenin inhibitors in combination with ICIs. In tumors in which T cells are abundant but immunosuppression by Tregs and TAMs are dominant, CTLA-4 blockade or VEGF/R blockade may help to deplete immunosuppressive cells and establish an immunocompetent TME. Lastly, current clinical trials in MSS CRC are focused on intractable stage IV patients with multiple metastases (especially liver metastasis). Such metastases might be an important reason why immunotherapy is ineffective in MSS CRC. This subgroup of patients may benefit less from immunotherapy because they have evolved strong immune-evasive mechanisms during disease progression and metastasis. Therefore, it is necessary to investigate the efficacy of immunotherapy in early-stage CRC patients with less immunosuppression. In particular, in high-risk stage II or stage III CRCs, because tumor burden is minimal after surgical resection, postoperative adjuvant immunotherapy may play a role in preventing the recurrence.

Along with the introduction of novel therapeutic agents, effective combination immunotherapy in MSS mCRC is expected to be established through future clinical trials.

## Figures and Tables

**Figure 1 cancers-13-04906-f001:**
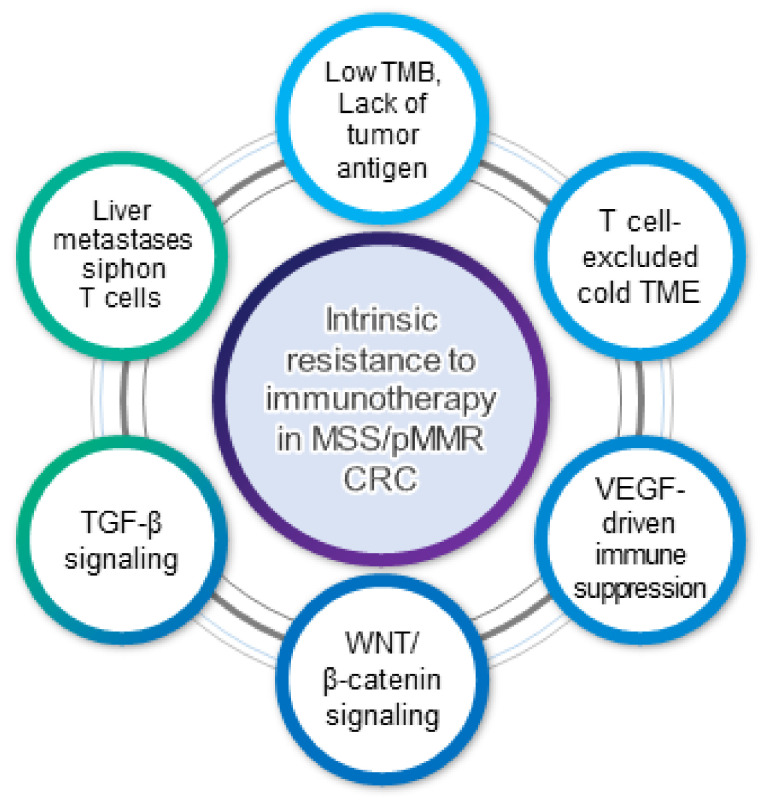
Mechanisms of resistance to immunotherapy in MSS/pMMR mCRC.

**Table 1 cancers-13-04906-t001:** Clinical trials using ICIs in patients with MSS/pMMR mCRCs.

Trial	Agent(s)	Target	Phase	Patients (n), MSS Rate	ORR	mPFS (mo)	mOS (mo)	Reference
KEYNOTE-016	Pem	PD-1	II	18, 100%	0%	2.2	5.0	[29]
KEYNOTE-028	Pem	PD-1	II	23, 96%	4%	1.8	5.3	[30]
CheckMate 142	Nivo* + Ipi	PD-1, CTLA-4	II	10, 100%	10%	2.3	11.5	[31]
	Nivo** + Ipi			10, 100%	0%	1.3	3.7	
CCTG CO.26	Durva + Treme	PD-L1, CTLA-4	II	119, 98%	1%	1.8	6.6	[32]
BACCI	Cape + Beva + Atezo	PD-L1, VEGF-A	II	82, 86%	9%	4.4	10.5	[33]
	Cape + Beva			46, 87%	4%	3.3	10.6	
REGONIVO	Nivo + Rego	PD-1, VEGFR	Ib	25, 96%	36%	7.9	12.3	[34]
Kim et al.	Nivo + Rego	PD-1, VEGFR	I	28, 100%	5%	4.3	11.0	[35]
REGOMUNE	Avel + Rego	PD-L1, VEGFR	II	48, 100%	0%	3.6	10.8	[36]
NCT0198889	Atezo + Cobi	PD-L1, MEK	I/Ib	84, 74%	8%	1.9	9.8	[37]
IMblaze370	Atezo + Cobi	PD-L1, MEK, VEGFR1/2/3	III	183, 93%	3%	1.9	8.9	[38]
	Atezo			90, 92%	2%	1.9	7.1	
	Rego			90, 100%	2%	2	8.5	

mPFS, median progression-free survival; Pem, pembrolizumab; PD-1, programmed cell death-1; Nivo, nivolumab; Ipi, ipilimumab; CTLA-4, cytotoxic T lymphocyte antigen-4; Durva, durvalumab; Treme, tremelimumab; PD-L1, programmed cell death ligand-1; Cape, capecitabine; Beva, bevacizumab; Atezo, atezolizumab; VEGF, vascular endothelial growth factor; Rego, regorafenib; VEGFR, vascular endothelial growth factor receptor; Cobi, cobimetinib; MEK, mitogen-activated protein kinase kinase; Nivo*, 1 mg/kg plus 3 mg/kg; Nivo**, 3 mg/kg plus 1 mg/kg.

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
