# Peer review of "Combination Immunotherapies to Overcome Intrinsic Resistance to Checkpoint Blockade in Microsatellite Stable Colorectal Cancer"

_cancers, 2021, doi:10.3390/cancers13194906_

Round 1

Reviewer 1 Report

This is an interesting review about combining ICI therapy with alternative approaches (including an additional ICI with an alternative mode of action) to overcome the resistance of MSS colorectal carcinomas ((CRCs) towards ICI monotherapy. The review reports on a variety of resistance mechanisms towards ICIs and approaches to (potentially) overcome them.

General: Disease/resistance mechanisms are addressed only marginally or not at all. E.g. through which mechanisms canonical WNT signalling attenuated recruitment of CD8+ cells to the tumor microenvironment. Shortly mentioning such mechanisms (e.g. “through modifying chemokine gradients” giving one or two up-to-date reference(s) would make the article more interesting for translational clinical scientists. Another example of an unclear mechanisms concerns Paxa-Vec.

Abstract

The abstracts refers to resistance mechanisms only. It would be more interesting to add to the Abstract at least the principal (not detailed) strategies (e.g. kinase inhibitors; radiotherapy) to overcome them.

Table 1 Footnote should explain the abbreviation mPFS. In the context of ICI trials, mPFS can denote “median PFS” or “modified PFS”. Please clarify.

Page 3: Please clarify abbreviation dMMR and pMMR the first time that they occur in the text

Page 4 TMB-H should be explicitly introduced as abbreviation in the text (the first time when it occurs in the text).

Page 4, line 128: this reviewer wonders whether this is a feed-forward loop (‘vicious cycle’) and not a feedback loop. Please check. 

Page 5: A sentence concerning the (presumed) mechanistic link between canonical WNT signalling in tumor cells and “exclusion of CD8+ T cells” from the TME should be added.

Page 6 Please extend on whether or not the inactivation of CD8+ T cells through liver metastasis is specific for CRC - or is also encountered in other cancers?

Page 5: Please give the function of regorafenib in a word in brackets when it appears for the first time in the text and not later in order to be consistent with other drugs, e.g. “cobimetinib (KEK inhibitor)” (line 178)

Page 6 Please check spelling/wording of sentence in lines 175-178. Otherwise the wording appears fine to me.

Page 6 Please explain meaning of FOLFOX when it occurs for the first time (line 216) and not later (line 219) in the text

Page 8 Please extend on the presumed (direct? Indirect?) mechanism how GM-CSF (Pexa-Vec)  “selectively kills tumor cells…” 

Page 9, CONCLUSION: This is largely a short summary and not a conclusion. In a good conclusion the authors should state which combination(s) in their opinion appear(s) as the most promising combination in the localized and metastatic setting. If it is premature to give such an opinion (or in case that combinations are generally not promising so far) this should be stated as well – and hopefully underpinned with a rational. A "perspective" might also improve the Conclusion.

Author Response

Reviewer 1

This is an interesting review about combining ICI therapy with alternative approaches (including an additional ICI with an alternative mode of action) to overcome the resistance of MSS colorectal carcinomas ((CRCs) towards ICI monotherapy. The review reports on a variety of resistance mechanisms towards ICIs and approaches to (potentially) overcome them.

General: Disease/resistance mechanisms are addressed only marginally or not at all. E.g. through which mechanisms canonical WNT signalling attenuated recruitment of CD8+ cells to the tumor microenvironment. Shortly mentioning such mechanisms (e.g. “through modifying chemokine gradients” giving one or two up-to-date reference(s) would make the article more interesting for translational clinical scientists. Another example of an unclear mechanisms concerns Paxa-Vec.

Thank you for this constructive comment. In the revised manuscript we have included more in detail descriptions on WNT/β-catenin signaling (line 123-134) and Pexa-vec (line 333-340).

Abstract

The abstracts refers to resistance mechanisms only. It would be more interesting to add to the Abstract at least the principal (not detailed) strategies (e.g. kinase inhibitors; radiotherapy) to overcome them.
Thank you for this valuable comment. We included strategies to overcome immunotherapy resistance in the revised abstract as follows:

“To overcome these resistance mechanisms, combination approaches using various agents, in-cluding STING agonists, MEK inhibitors, VEGF/R inhibitors, WNT/β-catenin inhibitors, oncolytic viruses, and chemo/radiotherapy, are actively ongoing. Preliminary evidence of efficacy for some has been shown in early clinical trials.”

Table 1 Footnote should explain the abbreviation mPFS. In the context of ICI trials, mPFS can denote “median PFS” or “modified PFS”. Please clarify.
We corrected accordingly in the revised manuscript (Line 81).

Page 3: Please clarify abbreviation dMMR and pMMR the first time that they occur in the text.

We corrected accordingly in the revised manuscript.

Page 4 TMB-H should be explicitly introduced as abbreviation in the text (the first time when it occurs in the text).

We changed “TMB-H” to “TMB-high” in the revised manuscript.

Page 4, line 128: this reviewer wonders whether this is a feed-forward loop (‘vicious cycle’) and not a feedback loop. Please check. 

As you suggested, “feed-forward loop” is exactly what we initially intended. We rephrased ‘vicious cycle’ to ‘feed-forward loop’ in the revised manuscript (line147-149).

Page 5: A sentence concerning the (presumed) mechanistic link between canonical WNT signaling in tumor cells and “exclusion of CD8+ T cells” from the TME should be added.
Thank you for this comment. In the revised manuscript (section 2.2), we have included more detailed mechanisms regarding WNT/β-catenin signaling and T cell exclusion as follows:

“β-catenin also suppressed the transcription of T-cell recruiting chemokine, CCL4, from dendritic cells and, thus, reduced the infiltration of CD8+ T cells within the TME in colon cancer and melanoma. Recent reports described that pharmacological inhibition of β-catenin could reactivate cancer immunity by increasing dendritic cells, upregulating CCL4, and promoting intratumoral infiltration of CD8+ T cells in various tumor models, including colon cancer. Moreover, β-catenin inhibition could strengthen the efficacy of PD-1 immune checkpoint blockade in multiple tumor models such as lung, breast, kidney, liver, and brain tumors. There is also crosstalk between WNT/β-catenin signaling and tumor-associated macrophages. Colon cancer cells promote IL-β secretion from macrophages. IL-β can prevent β-catenin degradation in colon cancer cells by phosphorylating GSK-3β.”

Page 6 Please extend on whether or not the inactivation of CD8+ T cells through liver metastasis is specific for CRC - or is also encountered in other cancers?

Recent studies reported the reduced efficacy of immune checkpoint blockades in various types of cancer with liver metastasis other than colorectal cancer, such as non-small cell lung cancer, melanoma, and gastric cancer. We have included this in the revised manuscript (section 2.5) as follows:

“In patients with advanced solid tumors treated with various immuno-oncologic agents in phase I trials, in which the most common histologies were melanoma and gastrointestinal cancers, liver metastases were associated with worse overall survival than the absence of liver metastases (median 8.1 vs. 21.9 months, P=0.017). Other studies have shown that patients with liver metastasis gain a reduced benefit from ICIs independent of other biomarkers in patients with melanoma and non-small cell lung cancer. Consistently, patients with advanced gastric cancer with hyperprogression of disease (HPD) after nivolumab treatment display more frequent liver metastases than those without HPD (77 vs. 41%, P=0.029). A recent study showed that liver metastases siphon activated CD8+ T cells from the systemic circulation and that hepatic myeloid cells induce T cell apoptosis through the Fas-FasL signaling pathway in preclinical models of colon cancer with liver metastasis, thus generating a systemic immune desert. Findings of another study suggested that liver metastasis induces tumor-specific immune suppression in distant tumors by activating Tregs and modulating CD11b+ monocytes in preclinical models of colon cancer.”

Page 5: Please give the function of regorafenib in a word in brackets when it appears for the first time in the text and not later in order to be consistent with other drugs, e.g. “cobimetinib (MEK inhibitor)” (line 178)
We have changed accordingly in the revised manuscript (line 260).

Page 6 Please check spelling/wording of sentence in lines 175-178. Otherwise the wording appears fine to me.
We have changed accordingly in the revised manuscript (line 207-209).

Page 6 Please explain meaning of FOLFOX when it occurs for the first time (line 216) and not later (line 219) in the text

We explained the meaning of FOLFOX (Folinic acid, fluorouracil, and oxaliplatin) in the revised manuscript (line 246).

Page 8 Please extend on the presumed (direct? Indirect?) mechanism how GM-CSF (Pexa-Vec) “selectively kills tumor cells…” 

We have included the mechanisms of Pexa-VEC in more detail in the revised manuscript as follows:

“Pexa-Vec (JX-594) is a genetically engineered oncolytic vaccinia virus in which the thymidine kinase (TK) gene has been deleted and the gene encoding human granulocyte macrophage colony-stimulating factor (GM-CSF) inserted. Because it lacks TK, it can replicate only in TK-overexpressing cancer cells, therefore showing high selectivity to cancer cells. Moreover, GM-CSF upregulation by Pexa-Vec facilitates the maturation and activation of dendritic cells within the TME, which enhances the cross-presentation of tumor-antigens and activation of T cells.”

Page 9, CONCLUSION: This is largely a short summary and not a conclusion. In a good conclusion the authors should state which combination(s) in their opinion appear(s) as the most promising combination in the localized and metastatic setting. If it is premature to give such an opinion (or in case that combinations are generally not promising so far) this should be stated as well – and hopefully underpinned with a rational. A "perspective" might also improve the Conclusion.

We really appreciate this valuable comment. We included our opinion on personalized combination approaches in patients with MSS CRC in the revised conclusion section as follows:

“Attempts to overcome the resistance of MSS CRC to ICIs are currently ongoing with the combined administration of various therapeutics targeting MEK, STING, VEGFR, or WNT, or using oncolytic viruses. Some of early clinical trials have revealed preliminary efficacy signals. However, the data are immature and need further validation. Because MSS mCRC is a cluster of immunologically heterogeneous tumors rather than a single disease entity, it is critical to precisely identify the main mechanism of resistance to immunotherapy in each patient and to apply personalized immunotherapeutic approaches accordingly. For example, in T-cell non-inflamed cold tumors, innate immune agonists such as STING agonists or oncolytic viruses may be effective in enhancing tumor immunogenicity and promoting intratumoral T cell infiltration. In addition, for cancers with very potent WNT/β-catenin activity, it may be better to use Wnt/β-catenin inhibitors in combination with ICIs. In tumors in which T cells are abundant but immunosuppression by Tregs and TAMs are dominant, CTLA-4 blockade or VEGF/R blockade may help to deplete immunosuppressive cells and establish an immunocompetent TME. Lastly, current clinical trials in MSS CRC are focused on intractable stage IV patients with multiple metastases (especially liver metastasis). Such metastases might be an important reason why immunotherapy is ineffective in MSS CRC. This subgroup of patients may benefit less from immunotherapy because they have evolved strong immune-evasive mechanisms during disease progression and metastasis. Therefore, it is necessary to investigate the efficacy of immunotherapy in early stage CRC patients with less immunosuppression. In particular, in high-risk stage II or stage III CRCs, because tumor burden is minimal after surgical resection, postoperative adjuvant immunotherapy may play a role in preventing the recurrence.”

Reviewer 2 Report

The review article is a focused expose looking at factors that influence resistance to checkpoint blockade in microsatellite stable cancers. The strength of the review is a timely presentation of recent studies in the literature in context of a major question in the field – what are the factors that render most CRC tumours intractable to immunotherapies.

I think that some of the factors presented by the authors that contribute to immunotherapy resistance should be put into context:

  • Active Wnt signaling leading to immune exclusion theory is based on correlative data, the only empirical evidence is a study using a mouse melanoma model.
  • The liver siphoning theory is an intriguing new result in the literature, arising from a single study and should be annotated as such. The authors give supporting lines of evidence from a number of studies – but should make clear that none of these provide a direct link.

Section 2. and 3. It’s not clear why these sections are separate with each of the associated subsections containing a number of redundant points. They should be combined.

Section 3.6 – A comment on what is to be learned about treating MSS CRCs from the anti-CTLA-4 studies would be useful.

For the studies that showed greater response rates in either MSS or MSI CRCs over the other, could the authors indicate where the data was controlled/stratified for stage of cancer development.

Line 338 – reference 110 – it’s not clear how this reference contributes to the point the authors are making in the paragraph.

For the conclusions paragraph, the authors state that the main mode of immunotherapy resistance should be identified for each patient and acted upon. I think this could be elaborated. For instance, could this understanding only be logistically reached post-resection – is the strategy here post-op treatment? Or perhaps a standard pre-op cocktail of inhibitors?

Line 206 – 5-fluorouracial

Author Response

Reviewer 2

The review article is a focused expose looking at factors that influence resistance to checkpoint blockade in microsatellite stable cancers. The strength of the review is a timely presentation of recent studies in the literature in context of a major question in the field – what are the factors that render most CRC tumours intractable to immunotherapies.

I think that some of the factors presented by the authors that contribute to immunotherapy resistance should be put into context:

  • Active Wnt signaling leading to immune exclusion theory is based on correlative data, the only empirical evidence is a study using a mouse melanoma model.
    Thank you for this comment. We have included more detailed mechanisms regarding WNT/β-catenin signaling and T cell exclusion in the revised manuscript as follows:

“β-catenin also suppressed the transcription of T-cell recruiting chemokine, CCL4, from dendritic cells and, thus, reduced the infiltration of CD8+ T cells within the TME in colon cancer and melanoma. Recent reports described that pharmacological inhibition of β-catenin could reactivate cancer immunity by increasing dendritic cells, upregulating CCL4, and promoting intratumoral infiltration of CD8+ T cells in various tumor models, including colon cancer. Moreover, β-catenin inhibition could strengthen the efficacy of PD-1 immune checkpoint blockade in multiple tumor models such as lung, breast, kidney, liver, and brain tumors. There is also crosstalk between WNT/β-catenin signaling and tumor-associated macrophages. Colon cancer cells promote IL-β secretion from macrophages. IL-β can prevent β-catenin degradation in colon cancer cells by phosphorylating GSK-3β.”

The liver siphoning theory is an intriguing new result in the literature, arising from a single study and should be annotated as such. The authors give supporting lines of evidence from a number of studies – but should make clear that none of these provide a direct link.

We really appreciate this comment. Recent studies reported the reduced efficacy of immune checkpoint blockade in various types of cancer with liver metastasis other than colorectal cancer, such as non-small cell lung cancer, melanoma, and gastric cancer. We have included more preclinical and clinical evidences in the revised manuscript as follows:

“In patients with advanced solid tumors treated with various immuno-oncologic agents in phase I trials, in which the most common histologies were melanoma and gastrointestinal cancers, liver metastases were associated with worse overall survival than the absence of liver metastases (median 8.1 vs. 21.9 months, P=0.017). Other studies have shown that patients with liver metastasis gain a reduced benefit from ICIs independent of other biomarkers in patients with melanoma and non-small cell lung cancer. Consistently, patients with advanced gastric cancer with hyperprogression of disease (HPD) after nivolumab treatment display more frequent liver metastases than those without HPD (77 vs. 41%, P=0.029). A recent study showed that liver metastases siphon activated CD8+ T cells from the systemic circulation and that hepatic myeloid cells induce T cell apoptosis through the Fas-FasL signaling pathway in preclinical models of colon cancer with liver metastasis, thus generating a systemic immune desert. Findings of another study suggested that liver metastasis induces tumor-specific immune suppression in distant tumors by activating Tregs and modulating CD11b+ monocytes in preclinical models of colon cancer.”

  • Section 2. and 3. It’s not clear why these sections are separate with each of the associated subsections containing a number of redundant points. They should be combined.

We subdivided sections 2 and 3 to improve readability. It would be appreciated if the reviewer would take this into consideration.

Section 3.6 – A comment on what is to be learned about treating MSS CRCs from the anti-CTLA-4 studies would be useful.

We have included a summary for this section in the revised manuscript as follows:

“Overall, there is limited evidence supporting the use of current-generation CTLA-4 antibodies in MSS/pMMR mCRC. However, next-generation CTLA-4 antibodies that can deplete intratumoral Tregs are expected to overcome immunotherapy resistance in patients with mCRC.”

For the studies that showed greater response rates in either MSS or MSI CRCs over the other, could the authors indicate where the data was controlled/stratified for stage of cancer development.

Most of studies are single-arm studies without control arm except BACCI and IMblaze370 trial. BACCI was a randomized, double-blind, phase II controlled trial and IMblaze370 was a randomized, open-label, phase III controlled trial. We have indicated this in the revised manuscript.

Moreover, clinical trials in this review article enrolled patients with metastatic CRC (mCRC) which is stage IV. We have indicated “mCRC” in clinical trial data throughout the revised manuscript.

Line 338 – reference 110 – it’s not clear how this reference contributes to the point the authors are making in the paragraph.

Reference 110 (Reference 123 in the revised manuscript) highlights the potential of immuno-radiotherapy in patients with MSS mCRC.

For the conclusions paragraph, the authors state that the main mode of immunotherapy resistance should be identified for each patient and acted upon. I think this could be elaborated. For instance, could this understanding only be logistically reached post-resection – is the strategy here post-op treatment? Or perhaps a standard pre-op cocktail of inhibitors?

Thank you for this comment. Current clinical trials in MSS CRC are focused on metastatic (stage IV) CRCs. However, pre-op or post-op immunotherapy may be beneficial in patients with early stage CRCs. We have included this in the revised conclusion paragraph as follows:

“Lastly, current clinical trials in MSS CRC are focused on intractable stage IV patients with multiple metastases (especially liver metastasis). Such metastases might be an important reason why immuno-therapy is ineffective in MSS CRC. This subgroup of patients may benefit less from immunotherapy because they have evolved strong immune-evasive mechanisms during disease progression and metastasis. Therefore, it is necessary to investigate the efficacy of immunotherapy in early stage CRC patients with less immuno-suppression. In particular, in high-risk stage II or stage III CRCs, because tumor burden is minimal after surgical resection, postoperative adjuvant immunotherapy may play a role in preventing the recurrence.”

Line 206 – 5-fluorouracial

Thank you very much. We corrected this typo.

Round 2

Reviewer 2 Report

Good rebuttal, thanks.